# VR-DiagNet: Medical Volumetric and Radiomic Diagnosis Networks with Interpretable Clinician-like Optimizing Visual Inspection

## ABSTRACT

Interpretable and robust medical diagnoses are essential traits for practicing clinicians. Most computer-augmented diagnostic systems suffer from three major problems: non-interpretability, limited modality analysis, and narrow focus. Existing frameworks can either deal with multimodality to some extent but suffer from non-interpretability or partially interpretable but provide a limited modality and multifaceted capabilities. Our work aims to integrate all these aspects in one complete framework to fully utilize the full spectrum of information offered by multiple modalities and facets. We propose our solution via our novel architecture **VR-DiagNet**, consisting of a planner and a classifier, optimized iteratively and cohesively. VR-DiagNet simulates the perceptual process of clinicians via the use of *volumetric* imaging information integrated with *radiomic* features modality; at the same time, it recreates human thought processes via a customized Monte Carlo Tree Search (MCTS) which constructs a volume-tailored experience tree to identify slices of interest (SoIs) in our multi-slice perception space. We conducted extensive experiments across two diagnostic tasks comprising six public medical volumetric benchmark datasets. Our findings showcase superior performance, as evidenced by heightened accuracy and area under the curve (AUC) metrics, reduced computational overhead, and expedited convergence while conclusively illustrating the immense value of integrating volumetric and radiomic modalities for our current problem setup.

## CCS CONCEPTS

• **Computing methodologies** → **Planning under uncertainty**.

## KEYWORDS

Medical Volume Diagnosis, Multimodality, Radiomics, Unsupervised Planning

## 1 INTRODUCTION

Accurate and robust diagnosis of diseases is a significant cornerstone of medical practice. In recent years, along with the development of computing power and algorithms, there has been increasing reliance on these systems to augment the work of clinicians. However, several problems with such systems hinder their continual integration into the medical workflow, including non-interpretability, limited modality analysis, and narrow focus. First, most diagnostic algorithms are black-boxes and, hence, non-interpretable. This presents a severe

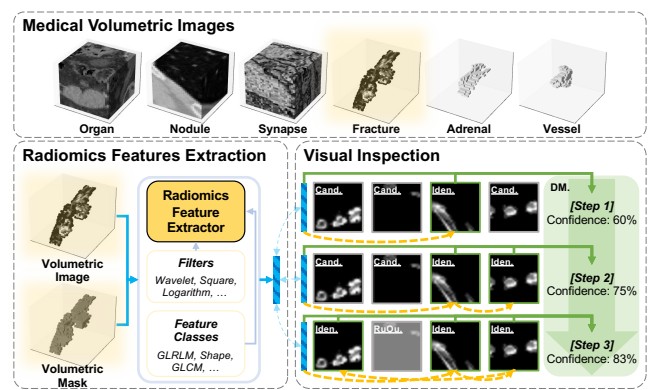

**Figure 1: This study endeavors to craft an interpretable diagnostic framework mirroring clinicians' visual inspection process. Leveraging radiomic and voxel features from different scales and computational methodologies bolsters the robustness of diagnostic procedures. Orange curved arrows indicate the step-wise formation of the decision path, while green right-angle arrows indicate the final decision formation. Cand. denotes a potential candidate, Iden. signifies an identified slice, RuOu. indicates a ruled-out slice not considered in decision-making.**

challenge for medical models because it renders their conclusions opaque and their inference processes cryptic and unintelligible [21]. This is unacceptable for conventional medical practice, which places high value on transparency in diagnosis and accountability of their results; this is, in fact, the hallmark of many mission-critical systems. In addition, most medically relevant learning models [32, 33] fail to adequately consider the multiple modalities of clinical data. With the increasing emphasis on personalized medicine [13] and its necessary prerequisites on highly customized features, it has become apparent that a single-modality approach towards healthcare has become woefully insufficient. Finally, most current learning models deployed can deal with a single [36] or, at most, a few aspects [37] of a medical problem. This is a significant obstacle to comprehensive and accurate diagnosis since, in most cases, medical conditions are complex and multifaceted, requiring the consideration and consolidation of disparate observations to reach the correct decision.

Recent advances in three-dimensional (3D) *volumetric imaging* have enabled imaging results to present a multi-sliced view of any medical condition, furnishing clinicians with a depth-rich source of visual information. Despite the clear advantages of volumetricity, most convolutional [25, 31, 35] and attention-based [11] frameworks extract visual features at the dense voxel-level, constructing a "global view" of the volume while overlooking its multifaceted nature. In addition to volumetricity, multimodality has yet to be comprehensively harnessed for increased diagnostic robustness and accuracy. Although the idea of multimodality is expected in the

medical scenario [1], very few works have attempted to leverage the large and diverse informational content of imaging feature space, or *radiomics*, for multimodal integration [20, 38]. With the advent of high-precision imaging equipment, radiomics have emerged as a novel imaging-based data characterization and feature identification system that enables extracting image features, or *phenotypes*, that comprehensively quantify the specifics of radiographic images. These features have been shown to offer valuable insights into intra- and inter-tumor heterogeneities [19]. Finally, interpretability [3] is an active and essential area of research in machine learning, especially in the medicine use scenario [3]. Moreover, it is apparent that interpretability and multimodal integration are often at odds with one another since multimodality (and especially integration thereof) often requires the design of a deep architecture, which is usually not amenable to interpretation [17, 18]. In this sense, the Monte Carlo Tree Search (MCTS) is a robust technique suitable for interpretable decision process planning and has been applied in the fields of Go[23], retrosynthetic planning[10], and games[15].

Having identified the critical bottlenecks in achieving interpretable, multimodal-multifaceted integration, we propose merging several essential paradigms to effectively unify these diverse yet crucial concepts. We illustrate our approach by considering two significant aspects of the problem: *perceptual* and *inferential*. At the perceptual level, we solve the problem of lack of multifacetedness and multimodality via the introduction of *volumetrics* and *radiomics*. Our approach combines descriptive features from both sources to optimize information extraction. Introducing our inference mechanism solves the problem of determining which features are descriptive. We have ensured that our inference logic is fully transparent and interpretable, thus ensuring the logical clarity of our approach.

To this end, we propose **VR-DiagNet**, namely Volumetric and Radiaomic Diagnosis Networks, which jointly incorporates volumetric imaging data and extracted radiomic features to encompass the multimodal integration for the diagnosis capacity enhancement. Moreover, the MCTS algorithm for identifiable inference processes is leveraged as a planning module. Our model comprises two primary multimodal constituents: the *class-agnostic planner* and the *classifier*. The planner identifies SoIs with the highest informativeness, enabling the alignment of the number of slices within each volume. Subsequently, sequence-aware strategy encoding is built upon the extracted slice-level local features and global radiomic features in the classifier, transformed into final diagnosis decision-making. The classifier has the same model structure as the planner, consisting of the Neighbour-aware Hierarchical Slice encoding (NHS) module for reading slice representation, the Sequence-aware Strategy Encoding (SSE) module for encoding the decision sequence, and the Decision Making (DM) layer that classifies the slice-level features output from SSE. This comprehensive integration of multimodality, multifacetness, and interpretability shows that our model improves upon baselines. In summary, our contributions are as follows:

- By identifying and surmounting key obstacles such as non-interpretability, narrow focus, and limited modality, we make decisive steps toward rectifying some of the severe issues hindering the effective application of deep learning in medical diagnosis;
- We present a framework for achieving multifaceted, multimodal medical diagnosis by designing the first learning framework, to the best of our knowledge, that emphasizes the effective interaction between radiomics and volumetric information, mediated by a robust decision-making process;
- We present a novel architecture consisting of an interpretable *planner* and a *multimodal, multifacetted* classifier, which closely mimics the inference strategies of human clinicians;
- We conduct extensive experiments, achieving state-of-the-art (SOTA) results while quantitatively and visually validating the effectiveness of integrating volumetric and radiomic features in the current multimodal tasks.

## 2 RELATED WORK

To offer a thorough overview, we explore specific technologies closely intertwined with this paper, including volumetric image diagnosis, medical radiomics analysis, and MTCS-based planning.

**Volumetric Image Diagnosis.** Deep learning has emerged as the dominant paradigm in medical image analysis (MIA) tasks in recent years. Convolution-based applications for medical volumetric images have adopted either 3D [25, 35] or 2.5D [31, 35] variants. These approaches, necessitating dense encoding, entail heightened computational costs and diminished data efficiency. Moreover, they are susceptible to over-fitting in scenarios with limited sample sizes. Contrarily, graph-based methodologies like GCN-CAP [14] and GSDG [4] leverage sparse encoding, reorganizing volume slices, effectively curbing computational complexity. This reduction is achieved by pre-extracting slice embeddings utilizing a frozen pre-trained Convolutional Neural Network (CNN) backbone during pre-processing. Although this approach significantly decreases input data density, it potentially suffers from domain gap issues introduced by out-of-domain pre-training. Jang et al. [11] propose a hybrid approach that merges Transformers with 2D and 3D CNNs, wherein the 3D CNN facilitates 3D representation learning while the 2D CNN utilizes pre-training weights from ImageNet [5] for 2D representation learning. However, the introduction of 3D models escalates the risk of over-fitting. Despite their different focuses, all the works above fail to utilize radiomic features, an under-utilized feature modality, and address how to model the visual inspection process of clinicians. Furthermore, techniques extending beyond raw imaging, such as quantitative analysis and feature extraction within the radiomics domain, need more emphasis in such endeavors despite their general efficacy in elucidating clinical insights.

**Medical Radiomics Analysis.** Radiomics features, with scales and calculation processes distinct from those of the imaging modality, complement the slice image modality and show promise in identifying slice- or volume-based biomarkers of therapy response. Reproducibility of radiomic features remains a challenge, addressed by initiatives like the Image Biomarker Standardization Initiative (IBSI) [41]. Tamal et al. [26] combine radiomics and machine learning for fast and accurate COVID-19 diagnosis from Chest X-ray images. Despite its good performance, classical machine learning remains dominant due to limited medical image samples and sparse radiomic data. Tanaka et al. [27] introduce a deep learning-based radiomics approach for early head and neck tumor regression prediction, outperforming traditional radiomics and clinical factors. However, its feature fusion stage precedes the machine learning decision layer, posing challenges in establishing deep-level complex

**Figure 2: The proposed architecture. Panel (a) demonstrates the formulation of a volume-specific experience tree utilizing the planner, depicted with three layers for clarity. The gray shadow represents the ancestral state of the state covered by the green shadow. Additionally, three SoIs are identified as a visual inspection strategy and highlighted in orange. In panel (b), the classifier extracts features from these SoIs and generates conditional predictions using an attention module with radiomic features and CLS embedding as the hierarchical priori. The static radiomic features are grouped and refined by another attention module, as shown in (c). Panel (d) showcases the design of the NHS module. Panel (e) presents the overall loss function used to train the classifier. "Att." stands for attention, "Lin." a linear layer, "RFR." radiomic feature refinement, "zpe" zero-centered position encoding and "⊕" element-wise addition. The NHS module utilizes a ResNet-18 backbone [9].**

correlations between the two modalities. Further exploration in multi-modal tasks is warranted to elucidate radiomics' underlying value. Ge et al. [8] demonstrate the effectiveness of extracting complementary radiomics features from diverse imaging modalities for identifying and differentiating kidney diseases. Vanguri et al. [29] integrate and analyze radiomic, pathological, and genomic features using machine learning algorithms in treatment response prediction. Deep Learning Radiomics (DLR) revolutionizes predictive modeling by extracting deep features. Zheng et al. [38] design a parallel model structure with ResNet-50 [9] base networks ingesting images of two modalities as dual-modal inputs. DLR extracts deep features from images while separating them from traditional radiomic features. Ning et al. [20] extract radiomics and deep features from MRI modalities to quantify global and local information using a kernel fusion-based SVM classifier for glioma grading. Our work aims to model the correlation between classical radiomic and deep features, enabling intermodality interaction and exploring inter-group interaction within radiomic features to adapt to deep feature distribution.

**MTCS-based Planning.** The emulation of the visual inspection process of clinicians is crucial for improving the interpretability of the model and inspiring the planner to align volumes with varying numbers of slices during the slice identification phase. Our methodology dissects volumetric images slice by slice, a tailored strategy for handling volumetric data. Furthermore, we integrate concepts

reminiscent of tree-based planning algorithms, such as MCTS, as observed in AlphaGo [23], and EG-MCTS [10], which addresses retrosynthetic planning for drug compounds. Based on the tasks at hand, we fill the gap in the works above. During the planning process, we suggest an additional data modality to help construct a visual inspection strategy similar to a clinician's. We explore a hierarchical priori design, consisting of volume-specific refined radiomic features constructed over static raw radiomic features and a CLS embedding within the Multi-Head Attention (MHA) mechanism [30]. The hierarchical design is anticipated to enhance diagnosis robustness owing to its multimodal and multigranular nature.

## 3 METHOD

### 3.1 Problem Statement

For accurate diagnostic purposes and by capturing the joint features between volumetric and radiomic features, our proposed multimodal method, VR-DiagNet, introduces a novel approach to medical image processing by integrating momentum planning (see §3.2) and volumetric & radiomic (VR) learning (see §3.3). The dataset $\left\{(\mathcal{X}_i, \mathbf{x}_{i,Rad}, y_i)\right\}_{i=1}^{N_D}$ comprises $N_D$ pairs of volume-radiomics-label. Each $\mathcal{X}_i$ consists of a sequence of $D_i$ gray-scale slices, with $D_i$ varying across volumes. $\mathbf{x}_{i,Rad}$ represents the radiomic feature vector, and $y_i$ denotes the corresponding volume-level label. The method

contains a planner $f_{PL}$ and a $C$-class classifier $f_{CL}$, optimized alternately over $N_{Ro}$ rounds. An overview of our method is illustrated in Figure 2.

## 3.2 Class-agnostic Momentum Planning

First, we introduce the planner, the first constituent in our method. To mimic human-like visual inspection akin to clinicians, we propose a computationally feasible planning methodology. In each round $r \in [1, \ldots, N_{Ro}]$, the planner constructs a volume-specific experience tree $\mathcal{T}_{\mathcal{X}_i}^{(r)}$ comprising $L$ layers (excluding the root node). This tree is built using voxels and radiomic features extracted from $\mathcal{X}_i$ and $\mathbf{x}_{i,Rad}$, enabling the selection of the most informative SoIs in a class-agnostic manner. This class-agnostic approach employs information metrics to determine the most task-relevant slices, eliminating the need for class labels and enabling generalization to any classification task. We formalize this process as a MDP characterized by $(\mathbb{S}, \mathbb{A}_s, \mathbb{P}_a, \mathbb{R})$:

(1) $\mathbb{S}$ represents the state space. Each slice $\mathcal{X}_{i,j}$ functions as a tree node. A state $s$ is constituted by a ordered node sequence $[\mathcal{X}_{i,RA}, \mathcal{X}_{i,[t(1),\ldots,t(l)]}]$, where $t(\cdot) \in [1, \ldots, D_i]$, and $1 \leq l \leq L$. We adopt a visual inspection **strategy** as a state $s$ in the rest of the paper. The node $\mathcal{X}_{i,RA}$ is a refined radiomic features learned from frozen and grouped sample-specific radiomic features $\mathbf{x}_{i,Rad}$ (see Eq. 3). The refinement aims to enhance radiomic complementarity to deep features. Details regarding radiomics feature extraction from volumes are elaborated in §4.3.1 and §4.3.2.

(2) $\mathbb{A}_s = \mathcal{X}_i \setminus s$ denotes candidate actions conditional on strategy $s$. An action $a \in \mathbb{A}_s$ selects a previously unvisited slice in $s$ and append it to the current strategy, assuming conditional independence regarding task-specific incremental information.

(3) $\mathbb{P}_a$ represents the transition function space. In our deterministic game setting, $P_a(s^P, s) = 1$ $(P_a(s^P, s) \in \mathbb{P}_a)$, indicating the transition probability from parent strategy $s^P$ to child strategy $s$ by taking action $a$.

(4) $\mathbb{R}$ denotes the reward space. $R(s) \in \mathbb{R}$ defines the average informativeness of strategy $s$, calculated using a value network consisting of SSE and DM modules (details in §3.3) based on normalized incremental informativeness (NII).

$$NII(s) = \begin{cases} NSE(s), & \text{if } \text{len}(s) = 1 \\ NMI(s), & \text{otherwise} \end{cases} \quad (1)$$

For a strategy with only one slice, NII is measured using a normalized negative variant of Shannon Entropy [22], $NSE(s) = \left( \log C + \sum_{c=1}^{C} \mathbf{p}_{s,c} \log (\mathbf{p}_{s,c} + \epsilon) \right) / \log C$, serving as a certainty metric. $\mathbf{p}_s \in [0,1]^C$ is the conditional probability distribution of the last slice in strategy $s$. For other strategies, NII is calculated using normalized mutual information: $NMI(s) = (\mathcal{H}^P - \mathcal{H}) / \sqrt{\mathcal{H}^P \mathcal{H}}$, where $\mathcal{H}$ and $\mathcal{H}^P$ are Shannon Entropy values for the strategy $s$ and its ancestor, respectively. This calculation also relies on conditional probability distributions.

We employ MCTS to solve the MDP above iteratively. The planner starts from the refined radiomics node and consists of four stages in each MCTS iteration:

(1) **Selection of the next slice:** Starting from the strategy $s^P$, we iteratively select and append the next slice to construct a new

---

**Algorithm 1** Pseudo-code for Strategy Reward Evaluation in MCTS

1: **Input**: Current visual inspection strategy $s$, action space $\mathbb{A}_s$.
2: **Output**: Reward $R(s)$.
3: **while** $\text{len}(s) < L$ and HasChild($s$) **do**
4: $\quad \mathbf{w}_{\mathbb{A}_s} \leftarrow \text{NII}(\mathbb{A}_s)$ $\qquad\qquad\qquad\qquad \triangleright$ *Eq. 1*
5: $\quad \mathcal{X}_{i,t(\text{len}(s)+1)} \leftarrow \text{WeightedSample}(\mathbb{A}_s, \text{weights} = \mathbf{w}_{\mathbb{A}_s})$
6: $\quad s \leftarrow s \cup \mathcal{X}_{i,t(\text{len}(s)+1)}$
7: **end while**
8: **return** $R(s)$

---

strategy $s$ using the Upper Confidence Bound (UCB) strategy [2], until reaching a leaf node or the maximum depth.

$$UCB1(s) = \underbrace{V(s)/N(s)}_{\textit{Exploration Term}} + c \underbrace{\sqrt{\log N(s^P)/N(s)}}_{\textit{Exploitation Term}} \quad (2)$$

Here, $V(s)$ tracks the accumulated $R(s)$, while $N(s)$ counts the exploration times of strategy $s$. The coefficient $c$ balances exploration versus exploitation.

(2) **Expansion of the visual inspection strategy.** If $N(s) \neq 0$, proceed to the next stage; otherwise, expand the experience tree by selecting an action $a$ that leads to an unvisited slice with the highest NII and append it to form the following strategy.

(3) **Evaluation of strategy reward.** Simulate on strategy $s$ to obtain reward $R(s)$. Termination occurs when the path depth reaches $L$ or a leaf node is reached. This stage is depicted in Alg. 1.

(4) **Metadata update of experience tree.** Update metadata of all ancestor strategies $s^P$ throughout the path: $N(s^P) \leftarrow N(s^P) + 1$ and $V(s^P) \leftarrow V(s^P) + R(s)$.

By leveraging slice informativeness informed by NII, the search direction of the planner aligns with the optimization direction of the classifier. Nonetheless, the extensive state space often necessitates the integration of MCTS with additional techniques to enhance efficiency and convergence [24]. The specific techniques we have employed are enumerated below.

*Efficiency optimization.* We employ the following strategies to enhance efficiency: 1) We adopt a layer-wise search strategy instead of exhaustively identifying all $L$ SoIs in consecutive MCTS iterations. Precisely, after every $N_{Mc}$ iterations, we pinpoint one child slice with the highest UCB score for the current initial slice. This selected child slice is the subsequent initial slice for the next $N_{Mc}$ iterations. Consequently, the iterative process involves $L \times N_{Mc}$ MCTS iterations for each volume in each round. 2) We project backbone embeddings of the SoIs into a low-dimensional feature space to reduce computational overhead. 3) We perform pre-computation of shared and parallelizable parts over SoIs within each volume on GPUs to alleviate the workload on CPUs.

*Momentum planner.* To ensure the stability of inter-round search directions, we propose a momentum-based initialization of $N(s)$ and $V(s)$: $N(\cdot)_{start}^{(r)} = m \times N(\cdot)_{end}^{(r-1)}$, $V(\cdot)_{start}^{(r)} = m \times V(\cdot)_{end}^{(r-1)}$, starting from the 3-rd round, where $m \in [0,1]$ is the momentum factor.

*Cold start.* At the beginning of the training, the planner does not adapt to medical images. Thus, we default to selecting the central $L$ slices from each volume and shuffling them to create ordered SoIs for warm-up training.

Thus far, the proposed planner yields a fine-grained knowledge sequence, denoted as $\mathcal{I}_i = [\mathcal{X}_{i,[t(1),...,t(L)]}]$, emulating a clinician-like visual inspection strategy, derived from $\mathcal{X}_i$. This sequence facilitates volume-level label assignment to identified SoIs. Furthermore, it is worth noting that by maintaining a consistent number of tree layers across volumes, the planner can achieve alignment regarding the number of SoIs across volumes of varying lengths.

## 3.3 Strategy-driven VR Learning

We partition the classifier into three constituents: **N**eighbour-aware **H**ierarchical **S**lice encoding (NHS) module, **S**equence-aware **S**trategy **E**ncoding (SSE) module, and **D**ecision **M**aking (DM) layer. The combination of SSE and DM is akin to the value network in AlphaGo [23], which can assess the informativeness of newly incoming slices. We denote the complete classifier as $f_{CL} = \{f_{NHS}, f_{SSE}, f_{DM}\}$, and its learning trajectory is directed by the supervised signal from the volume-level annotation. Below, we first elucidate the refinement of radiomic features and introduce the three modules.

*Radiomic feature refinement.* To align radiomic features with the same scale as the deep features, within each volume, we reorganize each raw radiomic features into corresponding groups, $\mathbf{x}_{i,Rad} \mapsto [\mathbf{g}_k]_{k=1}^{G}$, based on the radiomics taxonomy (see group names in §4.3.1). Here, $\mathbf{g}_k$ represents a subset of raw radiomic features, and $G$ denotes the number of groups. Each feature vector is then projected by a group-specific linear layer into the same feature space for subsequent self-attention interaction. Finally, the interacted features are mapped back and concatenated into the radiomics dimension:

$$\mathbf{x}_{i,Rad}^{int} = \|_{k=1}^{G} \text{Linear}_k^r \left( \text{MHA} \left( \text{Linear}_k^p (\mathbf{g}_k) \right) \right) \quad (3)$$

where $\|$ denotes the concatenation operation, $\text{Linear}_k^p$ and $\text{Linear}_k^r$ represent projection into uniform and back into raw radiomic feature sub-space, respectively. The resulting features are connected to the raw radiomic features by a shortcut implemented by a linear layer, as shown in panel (c), Figure 2. Layer Normalization follows their summation to obtain a refined radiomic feature vector $\mathbf{m}_{i,t(0)}$:

$$\mathbf{m}_{i,t(0)} = \text{LayerNorm} \left( \mathbf{x}_{i,Rad}^{int} \oplus \text{Linear} (\mathbf{x}_{i,Rad}) \right) \quad (4)$$

$\oplus$ indicates element-wise addition. Lastly, a linear layer is employed to map $\mathbf{m}_{i,t(0)}$ to the deep-feature dimension.

*Neighbour-aware hierarchical slice encoding.* We observe that neighboring slices offer moderate 3D information that is beneficial for current 3D tasks. Hence, we introduce the notion $\mathcal{X}_{i,t(l)}^{\mathcal{N}(n)}$ to represent the neighbor-aware version of the original $\mathcal{X}_{i,t(l)}$, covering $n$ neighboring slices in total. Here, $n$ is an odd number; thus, additional $\lfloor n/2 \rfloor$ slices before and after are concatenated with the target slice. For each augmented SoI, using ResNet-18 [9] as an example, a position-agnostic embedding $\mathbf{e}_{i,t(l)} = f_{NHS}(\mathcal{X}_{i,t(l)}^{\mathcal{N}(n)}) \in \mathbb{R}^{1024}$ is extracted from different encoder stages, specifically, the stem layer and four residual blocks, each followed by global average pooling. These embeddings are then concatenated to form a vector,

as shown in panel (d), Figure 2. Based on this, we compute an embedding sequence $\mathbf{e}_{i,\cdot} \in \mathbb{R}^{L \times 1024}$ by parallelly applying $f_{NHS}$ on $\mathcal{I}_i$. These embeddings are projected into a low-dimensional feature space $\mathbf{e}_{i,\cdot} = \text{ReLU} \left( \text{LayerNorm} \left( \text{Linear} (\mathbf{e}_{i,\cdot}) \right) \right) \in \mathbb{R}^{L \times rC}$ for computational efficiency, $r$ is scale coefficient.

*Sequence-aware strategy encoding.* We posit that the appearance of the next informative SoI is contingent upon all its predecessor SoIs within $\mathcal{I}_i$ and delineate the conditional distribution of $\mathcal{X}_{i,t(l)}$ as follows:

$$p \left( \mathcal{X}_{i,t(l)} \right) = p \left( \mathcal{X}_{i,t(l)} \mid \mathcal{X}_{i,[t(1),\cdots,t(l-1)]} \right) \quad (5)$$

We utilize MHA equipped with the proposed $\mathbf{m}_{i,t(0)}$ (Eq. 4), CLS embedding, dropout, and zero-centered position encoding $\mathbf{zpe}_{i,t(\cdot)}$ (details in §S.4, Supplementary Materials) to model this sequential relationship. For the (sub-) sequences of SoIs, we first calculate the updated conditional features $\mathbf{m}_{i,t(\cdot)} \in \mathbb{R}^{L \times rC}$ as follows:

$$\mathbf{m}_{i,t(\cdot)} = \text{MHA} \left( \mathbf{e}_{i,t(\cdot)} + \mathbf{zpe}_{i,t(\cdot)} \right) \quad (6)$$

In this context, MHA is performed slice-by-slice to guarantee that no descendant slice contributes to the computation of any ancestor slices. Each input embedding for the MHA corresponds to a node along the identified path within the experience tree.

*Decision making.* We employ a shared linear layer to obtain the position-wise logits of all SoIs:

$$logits_{i,t(\cdot)} = \text{Linear} \left( \text{ReLU} \left( \mathbf{m}_{i,t(\cdot)} \right) \right) \quad (7)$$

The volume-level prediction distribution $\hat{\mathbf{y}}_i \in [0,1]^{1 \times C}$ is Softmax (or Sigmoid for the binary classification setting) over the average of all slice logits:

$$\hat{\mathbf{y}}_i = \text{Softmax} \left( \text{Avg} \left( logits_{i,t(\cdot)}/\tau, dim = 0 \right) \right) \quad (8)$$

where $\tau$ represents temperature, however, for calculating the loss function, we opt for the slice-level distribution, i.e., without Avg above to get $\hat{\mathbf{y}}_{i,t(\cdot)} \in [0,1]^{L \times C}$.

*Classification loss function.* We formulate the likelihood of the identified SoIs as a joint probability:

$$\mathcal{L}_i(\theta) = \mathcal{L}_i(\theta; \mathcal{I}_i) = \prod_{l=1}^{L} \hat{\mathbf{y}}_{i,t(l)}(\mathcal{X}_{i,t(l)} | \mathcal{X}_{i,[t(1),\cdots,t(l-1)]}; \theta) \quad (9)$$

This likelihood is optimized by maximizing the log-likelihood, which is equivalently represented by minimizing cross-entropy (CE):

$$\mathcal{L}_{CL} = -\ln \mathcal{L}_i(\theta) = \frac{1}{L} \sum_{l=1}^{L} \ell_l = \frac{1}{L} \sum_{l=1}^{L} \text{CE} \left( \hat{\mathbf{y}}_{i,t(l)}, y_i \right) \quad (10)$$

*Penalty Term.* The penalty term is crucial for aligning the search direction of the planner with the optimization direction of the classifier and incentivizing the discovery of the most informative SoIs. For the $l$-th layer ($l \geq 2$), we define the penalty score between a strategy and its ancestor as:

$$\psi_l = \begin{cases} -\text{NMI}(s(l)), & \text{if NMI}(s(l)) < 0 \\ 0, & \text{otherwise} \end{cases} \quad (11)$$

Here, $s(l)$ represents a strategy corresponding to a $l$-layer tree path. Subsequently, the total penalty is calculated as follows:

$$\Psi = \frac{1}{L-1} \sum_{l=2}^{L} \psi_l \qquad (12)$$

The final loss function, balanced by $\lambda$, is then given by:

$$\mathcal{L} = \mathcal{L}_{CL} + \lambda \Psi \qquad (13)$$

*Training workflow.* The planner and classifier optimization alternate, mutually reinforcing, as shown in Alg. 2.

---

**Algorithm 2** Pseudo-code for Alternating Optimization

---

1: **Input**: Number of rounds $N_{Ro}$, number of volumes $N_D$, number of MCTS iterations to pinpoint one SoI $N_{Mc}$, training epochs in each round $N_{Ep}$, depth of the experience tree $L$.
2: **Output**: Planner $f_{PL}(;\theta)$, classifier $f_{CL}(;\theta)$
3: Initialize $f_{CL}$ with $\theta^{(0)}$;
4: **for** $r \leftarrow 1$ **to** $N_{Ro}$ **do**
5:    $D^{(r)} \leftarrow \{\}$         ▷ *Data set for the current round.*
6:    **for** $i \leftarrow 1$ **to** $N_D$ **do**
7:       $\mathcal{I}_i^{(r)} \leftarrow f_{PL}(\mathcal{X}_i, \mathbf{x}_{i,Rad}, N_{Mc}, L; \theta^{(r-1)})$   ▷ *Build Exp-tree and extract SoIs (§3.2).*
8:       $D^{(r)} \leftarrow D^{(r)} \cup (\mathcal{I}_i^{(r)}, \mathbf{x}_{i,Rad}, y_i)$
9:    **end for**
10:    $\theta^{(r)} \leftarrow \text{AdamW}\left(f_{CL}(;\theta^{(r-1)}), D^{(r)}, N_{Ep}\right)$ ▷ *Train classifier (§3.3).*
11: **end for**

---

## 4 EXPERIMENTS

### 4.1 Dataset and Pre-processing

The MedMNIST v2 dataset [35] offers a comprehensive benchmark featuring volumetric representations illustrating lesions across six distinct 3D diagnostic datasets with 9998 cases span modalities such as MRI, CT, and electron microscopy. The benchmark encompasses binary and multi-class classification tasks, with tensor dimensions of $28 \times 28 \times 28$ voxels per volume. Each $28 \times 28$ matrix along the first dimension is treated as a slice. We employ RandomResizedCrop() for data augmentation during the training phase. Value normalization is achieved using the z-score. These datasets are categorized into two problem domains following medical standards:

(1) *Task 1: anatomical structure and pathological condition analysis.* This stratification encompasses the classification of organ types (Organ3D), neuronal synapses (Synapse3D), and lung nodules (Nodule3D). They heavily rely on texture-based features for accurate classification. Details in [35].
(2) *Task 2: micro-structure classification.* This stratification analyzes fracture micro-structures (Fracture3D), intracranial aneurysms (Vessel3D), and adrenal gland morphology (Adrenal3D). They emphasize morphology-based characteristics, enabling the construction of fine-grained masks over variable-length volumes.

**Table 1: The classification of anatomical structures and pathological conditions was performed using VR-DiagNet, compared to approaches from various methodological families. Abbreviations include "Trans." for Transformer and "IL." for Incentive Learning. Results for AUC (↑) and ACC (↑) are provided.**

| Family | Method | Organ3D | | Nodule3D | | Synapse3D | |
|---|---|---|---|---|---|---|---|
| | | AUC | ACC | AUC | ACC | AUC | ACC |
| CNN | 2.5D$_{R18}$ [35] | 0.977 | 0.788 | 0.838 | 0.835 | 0.634 | 0.696 |
| | 3D$_{R18}$ [35] | 0.996 | 0.907 | 0.863 | 0.844 | 0.820 | 0.745 |
| | ACS$_{R18}$ [34] | 0.994 | 0.900 | 0.873 | 0.847 | 0.705 | 0.722 |
| | FRM$_{R18}$ [40] | 0.996 | 0.922 | 0.869 | 0.853 | 0.837 | 0.755 |
| Trans. | 2.5d Trans. [40] | 0.971 | 0.781 | 0.673 | 0.808 | 0.627 | 0.734 |
| IL. | C-Mixer [39] | 0.995 | 0.912 | **0.915** | 0.860 | 0.866 | 0.820 |
| Auto ML | auto-sklearn [7] | 0.977 | 0.814 | 0.914 | 0.874 | 0.631 | 0.730 |
| | AutoKeras [12] | 0.979 | 0.804 | 0.844 | 0.834 | 0.538 | 0.724 |
| Planning | **VR-DiagNet$_{R18}$** | **0.998** | **0.963** | 0.897 | **0.884** | **0.869** | **0.846** |

### 4.2 Settings

We delineate the shared hyper-parameters employed across all datasets. Specifically, we fix $N_{Ro} = 10$, $N_{Ep} = 20$, $\lambda = 0.1$, and set the batch size to 64. The neighboring size $(n)$ remains constant at 5 within $\mathcal{N}(n)$. We adopt $c = \sqrt{2}$ as the balancing factor for UCB and employ AdamW [16] as the optimizer. For a comprehensive overview of hyperparameter configurations, please consult Table S6, Supplementary Materials. The NHS module employs a ResNet-18 backbone [9] for comparison with prevalent convolutional models regarding model size, convergence speed, and computation cost due to the shared backbone. While our architecture could potentially utilize other visual backbones such as ViT [6], it falls beyond the scope of our current focus. Model selection hinges upon the validation set's accuracy (ACC) score. Each reported score reflects the average over three runs to maintain consistency with [35].

### 4.3 Results

*4.3.1 Anatomical Structure & Pathological Condition Analysis.* Extraction of coarse-grained radiomic features. During the feature extraction phase, we utilize a cubic binary mask, the same as the shape of the entire tensor, to capture 107 radiomic features per volume. This extraction process is facilitated by the PyRadiomics platform [28]. These features cover a broad spectrum, encompassing 18 *first-order* statistics, 16 *3D shape-based* descriptors, and various matrices including the *Gray Level Co-occurrence Matrix (GLCM)* comprising 24 features, *Gray Level Run Length Matrix (GLRLM)* with 16 features, *Gray Level Size Zone Matrix (GLSZM)* with 16 features, *Neighboring Gray Tone Difference Matrix (NGTDM)* with 5 features, and *Gray Level Dependence Matrix (GLDM)* with 14 features. Subsequently, z-score normalization is applied to standardize the radiomic features across volumes. Given the disparate contributions of different radiomic features to various tasks and lesions and recognizing the limited descriptive capability of cubic masks in delineating anatomical structures of interest, we adhere to established medical research practices [26]. Specifically, we utilize one-way ANOVA testing for each dataset to identify discriminative radiomic features exhibiting statistically significant differences

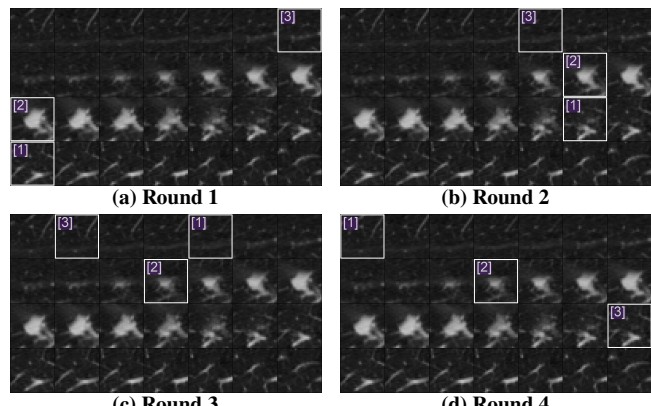

**Figure 3: Visualizing the planner's search process on a volume from Nodule3D across multiple rounds. Only the first four rounds are shown out of ten, with SoI indices along the tree path indicated in brackets. Complete results over ten rounds can be found in Figure S9, Supplementary Materials.**

($p < 0.05$) among different classes. This analytical process results in the following numbers of radiomic features for the respective datasets: Nodule3D: 62, Organ3D: 90, Fracture3D: 84, Adrenal3D: 68, Vessel3D: 83, and Synapse3D: 82. We quantitatively compared our proposed method's efficacy against SOTA techniques. The outcomes are summarized in Table 1. Our approach showcases either SOTA performance or comparability across all datasets, spanning diverse methodological families by attaining the highest average ACC and securing the second position in terms of average AUC.

*4.3.2 Micro-structure Classification.* Extraction of fine-grained radiomic features. Volumes from Fracture3D, Adrenal3D, and Vessel3D datasets possess a distinct pure black background. This characteristic enables the creation of a fine-grained mask instead of the coarse-grained cubic mask in §4.3.1. We intuitively argue that such a fine-grained mask could enhance the discriminability of extracted radiomic features and deep features derived from a multimodal model. Utilizing the PyRadiomics on the fine-grained binary mask and volumetric images, we obtained the following counts of radiomic features: Fracture3D: 85, Adrenal3D: 39, Vessel3D: 40. In contrast to task 1, this task accentuates the morphological characterization of the lesion. In addition to masking out background voxels for radiomic features, the exclusion of black slices results in **volumes of variable-length**. The datasets of such characteristics can be used to demonstrate our method's capability to process volumes of varying lengths. We benchmark our approach with and without variable-length volumes against the SOTA, as illustrated in Table 2. The elimination of black slices (the last row) restricts the search space, resulting in higher efficiency in lesion retrieval, thereby enhancing diagnostic accuracy relative to searching within 28 slices (penultimate row).

*4.3.3 Influence of Radiomic Features in Multimodal Contexts.* We proceeded to conduct experiments involving various designs of priori embeddings and present the results in Table 3. These results unequivocally demonstrate an enhancement in model performance attributable to integrating radiomic features, thereby validating their utility in such tasks. The CLS embedding can be conceptualized as a

**Table 2: VR-DiagNet was evaluated against SOTA methods using three datasets for micro-structure analysis, comprising samples with a black background and distinct morphological structures. The superscript ∗ indicates training on variable-length volumes excluding black slices, limiting candidate slice search.**

| Family | Method | Fracture3D (3) AUC | Fracture3D (3) ACC | Adrenal3D (2) AUC | Adrenal3D (2) ACC | Vessel3D (2) AUC | Vessel3D (2) ACC |
|---|---|---|---|---|---|---|---|
| CNN | 2.5D$_{R18}$ [35] | 0.587 | 0.451 | 0.718 | 0.772 | 0.748 | 0.846 |
| | 3D$_{R18}$ [35] | 0.712 | 0.508 | 0.827 | 0.721 | 0.874 | 0.877 |
| | ACS$_{R18}$ [34] | 0.714 | 0.497 | 0.839 | 0.754 | 0.930 | 0.928 |
| | FRM$_{R18}$ [40] | 0.588 | 0.433 | 0.870 | 0.819 | 0.931 | 0.918 |
| Trans. | 2.5d Trans. [40] | 0.583 | 0.402 | 0.657 | 0.768 | 0.598 | 0.887 |
| IL | C-Mixer [39] | **0.729** | **0.660** | **0.969** | 0.801 | 0.932 | 0.940 |
| Auto ML | auto-sklearn [7] | 0.628 | 0.453 | 0.828 | 0.802 | 0.910 | 0.915 |
| | AutoKeras [12] | 0.642 | 0.458 | 0.804 | 0.705 | 0.773 | 0.894 |
| Planning | **VR-DiagNet**$_{R18}$ | 0.673 | 0.527 | 0.854 | 0.805 | 0.946 | 0.940 |
| | **VR-DiagNet**$^*_{R18}$ | 0.708 | 0.561 | 0.868 | **0.828** | **0.950** | **0.953** |

**Table 3: Different implementations of priori embeddings were tested, and the averaged AUC (↑) and ACC (↑) were reported over six datasets. Two key implementations were examined. Volume-level: trainable refinement over static radiomic features. Task-level: a dataset-level trainable CLS embedding.**

| Methods | Volume-level | Task-level | AUC | ACC |
|---|---|---|---|---|
| Naïve CLS embedding [30] | × | √ | 0.853 | 0.818 |
| VR-DiagNet (Ours) | √ | × | 0.867 | 0.821 |
| | √ | √ | **0.870** | **0.826** |

task-level trainable priori, as analyzed in Related Works (§2), and the refined radiomic features function as a volume-level trainable priori. We employed a hierarchical priori design comprising both types of priori embeddings in subsequent experiments.

*4.3.4 Visualization.* The normalized feature strength maps of radiomic features before and after refinement across six datasets are presented in Figure S2 to S7, Supplementary Materials. Notably, it is evident that our adopted multimodal clinician-like diagnostic approach altered the distribution weights of features. Given our primary focus on computer algorithm design, we refrained from delving deeper into the associations between distinct radiomic features and various tissues or diseases, a pursuit commonly undertaken in clinical research. Instead, we treated the refined radiomic features post-feature selection as a collective feature vector. We hope these insights will benefit clinical research endeavors.

We illustrate samples from two datasets: one with a black background, as illustrated in Figure 4, and the other lacking such background, as demonstrated in Figure 3. The planner gradually emphasizes the identification of the most informative SoIs as the learning iterates. Furthermore, these delineated SoIs stimulate clinicians' visual inspection process, thus bolstering the interpretability of the deep learning model. Figure S8 to S13, Supplementary Materials provides comprehensive visualizations across six datasets.

*4.3.5 Ablation Study.* We assume that the radiomic features possess an inherently "high-level" character, as they demarcate the 3D lesion

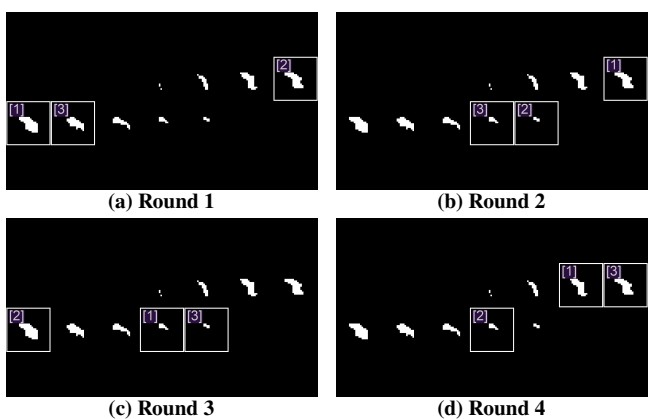

**(a) Round 1**  **(b) Round 2**

**(c) Round 3**  **(d) Round 4**

**Figure 4: Visualization of the planner's search process on a volume from Adrenal3D over the first four rounds is presented. Please refer to Figure S11 in the Supplementary Materials for the complete results over ten rounds.**

**Table 4: The choice of types of radiomic features. ACC (↑) across six datasets are reported.**

| Fea. Type | Org. | Nod. | Fra. | Adr. | Ves. | Syn. | Avg. |
|---|---|---|---|---|---|---|---|
| Static | 0.944 | **0.884** | 0.549 | **0.818** | **0.943** | 0.819 | 0.826 |
| Refined | **0.958** | 0.865 | **0.577** | 0.809 | 0.937 | **0.836** | **0.830** |

**Table 5: Comparison of diagnostic performances after radiomic feature extraction across six datasets using different mask granularities. AUC (↑) and ACC (↑) are reported.**

| Granularity | Fracture3D | | Adrenal3D | | Vessel3D | | Average | |
|---|---|---|---|---|---|---|---|---|
| | AUC | ACC | AUC | ACC | AUC | ACC | AUC | ACC |
| Coarse-grained | 0.690 | 0.540 | **0.860** | **0.818** | 0.915 | 0.915 | 0.822 | 0.758 |
| Fine-grained | **0.718** | **0.549** | 0.846 | **0.818** | **0.953** | **0.942** | **0.839** | **0.770** |

of interest from a comprehensive 3D standpoint, typically integrated directly into traditional machine learning classifiers. Consequently, they may not be optimally conducive to deep learning models. We scrutinize this assumption by contrasting static raw radiomic features with trainable refined radiomic features, as delineated in Table 4. The findings suggest that each feature type possesses unique merits across different datasets, with the static scheme yielding a marginally higher average ACC. Nevertheless, we have opted for the trainable design approach to delve deeper into the influence of multimodal deep models on radiomic features.

From the perspective of varying granularity of masks used in extracting radiomic features, Table 5 examines the discriminability of deep features when incorporating radiomic features of different granularity as the second modality. It is observed that the utilization of fine-grained masks leads to enhanced macro-averaged ACC and AUC metrics, highlighting the advantageous impact of finer masks. Additionally, we present outcomes utilizing classical machine learning models, specifically an SVM classifier with an RBF kernel, a methodology commonly adopted in medical research as elucidated

**Table 6: Analysis of parameter count (↓) and training-time FLOPS (↓) on a single volume across different methodologies.**

| Methods | # Params (M) | FLOPS (B) |
|---|---|---|
| $3D_{R18}$ [35] | 33.15 | 106.42 |
| $2.5D_{R18}$ [35] | **11.17** | 35.64 |
| $ACS_{R18}$ [34] | **11.17** | 35.64 |
| $f_{CL}$ w/ $L = 6$, $r = 8$ (Organ3D) | 11.36 | 16.46 |
| $f_{CL}$ w/ $L = 3$, $r = 4$ (Nodule3D) | 11.18 | 13.71 |
| $f_{CL}$ w/ $L = 7$, $r = 16$ (Synapse3D) | 11.20 | 19.19 |
| $f_{CL}$ w/ $L = 3$, $r = 64$ (Fracture3D) | 11.72 | 8.24 |
| $f_{CL}$ w/ $L = 3$, $r = 4$ (Adrenal3D) | 11.18 | 13.71 |
| $f_{CL}$ w/ $L = 3$, $r = 8$ (Vessel3D) | 11.18 | **8.22** |

in [26]. The classification scores are presented in Table S7, Supplementary Materials, provide insights into the discriminative capacity of deep learning-refined radiomic features compared to their static counterparts. Encouragingly, an overall performance enhancement is observed, suggesting a potential medical research advancement.

We analyze the model size and computational overhead during training, juxtaposing our classifier with competing approaches, as detailed in Table 6. Specifically, we choose to compare with R18+3D [35], R18+2.5D [35], and R18+ACS [34], as they employ the same backbone architecture. Notably, ACS [34] distinguishes itself by achieving comparable accuracy with reduced computational complexity compared to the naïve 3D scheme. Our analysis reveals a significant reduction in floating-point operations per second (FLOPS) in VR-DiagNet, attributable to the downsampling of slices, thus alleviating computational burdens. In contrast, competitors operate on all 28 slices, resulting in higher FLOPS. Remarkably, our approach maintains a comparable model size to the 2.5D and ACS counterparts. Moreover, we conduct a convergence analysis by training R18+ACS [34] for 200 epochs to align with the total epochs specified in our methodology. The results, as depicted in Figure S1, Supplementary Materials, showcase our method's superior convergence characteristics across all six datasets.

More extensive ablation study results can be found in §S.3, Supplementary Materials.

## 5 CONCLUSIONS

Modern medical automation needs to improve in several aspects: the black-box nature of the learning models, the single-faceted nature of their inference processes, and the limited ability to handle multimodal information. Our approach, VR-DiagNet, pioneers a model that integrates volumetric and radiomic features, from which information is extracted via an interpretable reasoning framework. This methodology consists of two crucial modules: a planner and a classifier. By incorporating clinicians' diagnostic reasoning into the framework of MDP, the planner utilizes MCTS to extract annotated SoIs, which serve as training data for the classifier. The introduction of class-agnostic information metrics ensures coherent optimization across both modules. We achieve SOTA performance in five of six diagnostic datasets for addressing volumetric tasks in MIA. With VR-DiagNet, we show that robust and accurate deep-learning models for medical use are entirely feasible.

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
