# OpenReview forum: "VR-DiagNet: Medical Volumetric and Radiomic Diagnosis Networks with Interpretable Clinician-like Optimizing Visual Inspection"
_acmmm.org/ACMMM/2024/Conference — MM2024 Poster_

### Official Review · Reviewer_NVqN · 2024-05-19

**Rating:** 6
**Confidence:** 3

**Summary:**

The manuscript addresses critical challenges in the application of deep learning to medical diagnosis, emphasizing the need for interpretability, multimodal integration, and comprehensive analysis. In response, they propose VR-DiagNet, a novel framework combining volumetric imaging and radiomic features to enhance diagnostic capabilities. This approach leverages Monte Carlo Tree Search (MCTS) for interpretable decision-making and integrates descriptive features from multiple sources for optimal information extraction. The model comprises a class-agnostic planner and a classifier, both designed to handle multifaceted and multimodal data, ensuring transparency and logical clarity in inference. The manuscript makes notable contributions by presenting a comprehensive and interpretable framework that significantly improves upon existing baselines in medical diagnosis tasks. Extensive experiments demonstrate the model's effectiveness, setting a new standard for integrating volumetric and radiomic features in multimodal medical diagnosis. This work represents a substantial advancement in the field, addressing key limitations and offering a robust solution for real-world clinical applications.

**Strengths:**

The approach proposed in the paper is solving a problem of significant importance. They have proposed a novel approach and demonstrated state-of-the-art results on multiple open datasets. They have described their approach, datasets, implementation details, evaluation metrics and results very well.

**Limitations:**

I don't see any major concerns with the paper. Although, I am a bit unsure how this paper is directly related to multimedia but the approach discussed in the paper can certainly be extended to multimedia use cases as well and hence can act as a source of inspiration which can drive further research in the field of multimedia.

**Suitability:**

2

---

### Official Review · Reviewer_Cg26 · 2024-05-24

**Rating:** 3
**Confidence:** 4

**Summary:**

The paper introduces an innovative diagnostic framework, VR-DiagNet, designed to enhance the accuracy and efficiency of clinical diagnostics using volumetric imaging data. The core components of the framework are a planner and a classifier, which are optimized in a cohesive and iterative manner. This framework simulates the perceptual processes of clinicians by integrating volumetric imaging information with radiomic features, allowing for a more comprehensive analysis.

**Strengths:**

VR-DiagNet enhances the interpretability of deep learning models by integrating volumetric imaging information with radiomic features, which clinicians can more easily relate to and understand.'

The framework overcomes the narrow focus of traditional models by utilizing a customized Monte Carlo Tree Search (MCTS) to construct a volume-tailored experience tree.

The proposed framework is specifically designed to achieve a multifaceted, multimodal approach to medical diagnosis. This is a pioneering step in the field, offering a cohesive system that integrates various types of medical data and simulates the perceptual and cognitive processes of clinicians.

**Limitations:**

The low resolution (28x28x28) of the MedMNIST images may not adequately represent the complexity and detail present in higher-resolution medical images typically used in clinical settings. As a result, comparing VR-DiagNet's performance with state-of-the-art (SOTA) methods using such a dataset could lead to biased or misleading conclusions about its efficacy and superiority.

As observed in Table 1 of the paper, VR-DiagNet did not achieve the best Area Under the Curve (AUC) for all three structures analyzed. Similarly, Table 2 shows that the proposed method did not produce the best results across the three structures evaluated. These results indicate that while VR-DiagNet shows promise, it does not consistently outperform existing methods on the chosen dataset, highlighting potential limitations in its current implementation or optimization.

To fully validate the effectiveness and applicability of VR-DiagNet, it is crucial to test the framework on datasets with higher image resolutions. For instance, using datasets like ChestX-ray14 [1], which features images with resolutions around 1024x1024, would provide a more comprehensive evaluation. Higher resolution images better mimic real-world scenarios, allowing for a more accurate assessment of the method's performance and its potential advantages over current SOTA techniques.

References:
[1] X. Wang, Y. Peng, L. Lu, Z. Lu, M. Bagheri, and R. M. Summers, "ChestX-ray8: Hospital-scale chest X-ray database and benchmarks on weakly-supervised classification and localization of common thorax diseases," in *Proc. IEEE Conf. Comput. Vis. Pattern Recognit.*, 2017, pp. 3462-3471.

**Suitability:**

3

---

### Official Review · Reviewer_hxfQ · 2024-05-24

**Rating:** 4
**Confidence:** 1

**Summary:**

The paper proposes a method called VR-DiagNet for medical volumetric and radiomic diagnosis. The method integrates volumetric imaging information with radiomic features and utilizes a planner and a classifier to simulate the perceptual process of clinicians. The planner uses a Monte Carlo Tree Search (MCTS) to identify slices of interest **Generated by ChatGPT, no copying allowed!** in the volume, while the classifier incorporates radiomic features and deep features to make the final diagnosis decision. The authors conducted extensive experiments on six public medical volumetric benchmark datasets and achieved superior performance compared to baselines.

**Strengths:**

1) The paper addresses the important problem of interpretable and robust medical diagnoses, which are essential for practicing clinicians.
2) The proposed VR-DiagNet method integrates volumetric imaging information and radiomic features, which allows for a more comprehensive and accurate diagnosis.
3) The use of MCTS in the planner module provides an interpretable decision-making process, which enhances the transparency of the model.

**Limitations:**

1) The evaluation of the proposed method is limited to 2 tasks. It would be beneficial to compare the performance of VR-DiagNet with other state-of-the-art methods on a wider range of datasets.
2) The paper could benefit from a more thorough discussion of the limitations and potential future directions of the proposed method.

**Suitability:**

3

---

### Meta-Review · Area_Chair_XWaJ · 2024-06-25

**Recommendation:** Accept (Poster)
**Confidence:** 5

**Metareview:**

All reviewers agreed on the submission and responses as provided.